# Differentially Expressed Genes Involved in Primary Resistance to Immunotherapy in Patients with Advanced-Stage Pulmonary Cancer

**DOI:** 10.3390/ijms25042048

**Published:** 2024-02-08

**Authors:** Luis Miguel Chinchilla-Tábora, Juan Carlos Montero, Luis Antonio Corchete, Idalia González-Morais, Edel del Barco Morillo, Alejandro Olivares-Hernández, Marta Rodríguez González, José María Sayagués, María Dolores Ludeña

**Affiliations:** 1Department of Pathology, Institute for Biomedical Research of Salamanca (IBSAL), University Hospital of Salamanca, University of Salamanca, 37007 Salamanca, Spain; lmchinchilla@saludcastillayleon.es (L.M.C.-T.); jcmon@usal.es (J.C.M.); idalia_dsks@hotmail.com (I.G.-M.); martarodriguez@saludcastillayleon.es (M.R.G.); 2Biomedical Research Networking Centers-Oncology (CIBERONC), 28029 Madrid, Spain; 3Department of Medicine, Harvard Medical School, Boston, MA 02214, USA; lacorsan@usal.es; 4Department of Medical Oncology, Institute for Biomedical Research of Salamanca (IBSAL), University Hospital of Salamanca, University of Salamanca, 37007 Salamanca, Spain; ebarco@saludcastillayleon.es (E.d.B.M.); aolivares@saludcastillayleon.es (A.O.-H.)

**Keywords:** anti-PD1/PD-L1, immunotherapy, nivolumab, NSCLC, GEP

## Abstract

In the last few years, nivolumab has become the standard of care for advanced-stage lung cancer patients. Unfortunately, up to 60% of patients do not respond to this treatment. In our study, we identified variations in gene expression related to primary resistance to immunotherapy. Bronchoscopy biopsies were obtained from advanced non-small cell lung cancer (NSCLC) patients previously characterized as responders or non-responders after nivolumab treatment. Ten tumor biopsies (from three responders and seven non-responders) were analyzed by the differential expression of 760 genes using the NanoString nCounter platform. These genes are known to be involved in the response to anti-PD1/PD-L1 therapy. All the patients were treated with nivolumab. Examining the dysregulated expression of 24 genes made it possible to predict the response to nivolumab treatment. Supervised analysis of the gene expression profile (GEP) revealed that responder patients had significantly higher levels of expression of *CXCL11, NT5E, KLRK1, CD3G, GZMA, IDO1, LCK, CXCL9, GNLY, ITGAL, HLA-DRB1, CXCR6, IFNG, CD8A, ITK, B2M, HLA-B,* and *HLA-A* than did non-responder patients. In contrast, *PNOC, CD19, TP73, ARG1, FCRL2,* and *PTGER1* genes had significantly lower expression levels than non-responder patients. These findings were validated as predictive biomarkers in an independent series of 201 patients treated with nivolumab (22 hepatocellular carcinomas, 14 non-squamous cell lung carcinomas, 5 head and neck squamous cell carcinomas, 1 ureter/renal pelvis carcinoma, 120 melanomas, 4 bladder carcinomas, 31 renal cell carcinomas, and 4 squamous cell lung carcinomas). ROC curve analysis showed that the expression levels of *ITK, NT5E, ITGAL*, and *CD8A* were the best predictors of response to nivolumab. Further, 13/24 genes showed an adverse impact on overall survival (OS) in an independent, large series of patients with NSCLC (2166 cases). In summary, we found a strong association between the global GEP of advanced NSCLC and the response to nivolumab. The classification of NSCLC patients based on GEP enabled us to identify those patients who genuinely benefited from treatment with immune checkpoint inhibitors (ICIs). We also demonstrated that abnormal expression of most of the markers comprising the genomic signature has an adverse influence on OS, making them significant markers for therapeutic decision-making. Additional prospective studies in larger series of patients are required to confirm the clinical utility of these biomarkers.

## 1. Introduction

Lung cancer is the leading cause of cancer-related death worldwide, with morbidity and mortality rates as high as 18%, corresponding to 1.8 million deaths per year [1]. Eighty-five percent of lung neoplasms fall within the non-small cell lung carcinoma (NSCLC) group. The two major histological types in this group of neoplasms are adenocarcinoma (ADC) and squamous cell carcinoma (SCC).

In recent decades, much progress has been made to further our knowledge of the diagnostic and therapeutic approach to cancer in general and lung cancer in particular [2]. In this regard, significant information has been gathered about predictive biomarkers in NSCLC [3]. However, most of them have been identified in the ADC histological subtype. In fact, the ADC genome has been fully characterized [4]. In contrast, further research into the genetic/genomic alterations of SCC is needed to advance our understanding of its biology, genomics, and susceptibility to new therapeutic targets [5]. In this context, alterations at the levels of the epidermal growth factor receptor (*EGFR*) and anaplastic lymphoma kinase (*ALK*) rearrangements deserve special attention because of their therapeutic significance. It is estimated that one in three patients with ADC histology benefit from this type of treatment—in these patients, it prevents early relapses and improves overall patient survival [6]. However, the application of molecular therapies targeting SCC in the clinical setting has been more limited, thereby making it a focus for research [7]. For this histological subtype, immunotherapy currently offers new hope, having been successfully applied in some patients [8]. In 2015, the Food and Drug Administration (FDA) approved pembrolizumab and nivolumab as second-line treatments for NSCLC [9]. Immune checkpoint inhibitors (ICIs) targeting PD-L1, such as nivolumab, have dramatically improved the treatment landscape for advanced non-small cell lung cancer (NSCLC). This has resulted in radical improvements in patient survival and quality of life [10]. However, although patients on ICI therapy can achieve long-term overall survival (OS), the mechanisms underlying resistance and/or sensitivity to PD-L1/PD-1 blockade are currently poorly understood, driving an urgent need to identify new biomarkers capable of accurately predicting the response to immunotherapy in these patients [11]. At present, the gold standard predictive biomarker of response to immunotherapy, the PD-L1 antigen studied by immunohistochemistry techniques, is not closely associated with treatment efficacy [12,13]. Thus, it is common to find tumors with a high level of PD-L1 expression that do not respond to treatment. Conversely, some patients with tumors lacking PD-L1 expression achieve complete responses after treatment with nivolumab [14], as we also noted in our study. This may be due to the great diversity of PD-L1 clones studied [15] and to patients’ great heterogeneity in terms of histology, molecular characteristics, therapeutic responses, and prognoses [16,17].

The gene expression profile (GEP) is an interesting tool that can be used to identify new genetic markers, allowing the simultaneous study of several thousand specific genes. In the specific case of breast cancer, tools based on GEP analysis, such as PAM50^®^ (Prosigna; NanoString Technologies, Seattle, WA, USA) and Oncotype DX^®^ (RS; Genomic Health, Redwood City, CA, USA), have been routinely used in clinical practice as fundamental predictors of early relapse, making them of great value for identifying biomarkers associated with the prognosis and treatment of patients with different types of neoplasm [18,19,20,21]. The identification of these molecules will allow a faster and more accurate decision-making protocol to be developed for the personalized treatment of NSCLC.

In the present study, we used the NanoString nCounter platform (Seattle, WA, USA) to analyze the differential expression of 760 genes that are involved in the complex interaction among the tumor, microenvironment, and immune response, and between NSCLC patients who respond and do not respond to nivolumab treatment. It is hoped that this will provide insight into the mechanisms of immune evasion and/or resistance with this specific drug. Overall, our study revealed a genomic signature comprising 24 genes that a multivariate analysis showed to be closely correlated with the response to ICI treatment and with disease prognosis. This information improves our understanding of the genomic landscape of ICI resistance in NSCLC patients.

## 2. Results

### 2.1. Gene Expression Profile (GEP) of Tumor Tissue of Patients with NSCLC after Nivolumab Treatment

We analyzed the differential expression of 760 genes involved in the response to anti-PD-L1 therapy using the NanoString nCounter platform (Seattle, WA, USA) in responder (*n* = 3) and non-responder (n = 7) patients, all of whom were treated homogeneously with nivolumab (Table 1). Supervised analysis of the GEP revealed that responder patients consistently contained the same dysregulated transcripts as those observed in non-responders (Figure 1A). However, some transcripts (n = 24) turned out to be specifically deregulated in responder patients, but not in non-responder patients, reflecting a greater genomic instability in patients who achieved complete remission. Notably, responder patients showed expression levels of CXCL11, NT5E, KLRK1, CD3G, GZMA, IDO1, LCK, CXCL9, GNLY, ITGAL, HLA-DRB1, CXCR6, IFNG, CD8A, ITK, B2M, HLA-B, and HLA-A that were significantly higher than in non-responder patients (Table 2; *p* < 0.05). In contrast, PNOC, CD19, TP73, ARG1, FCRL2, and PTGER1 genes had significantly lower expression levels in responder than in non-responder patients (Figure 1B).

### 2.2. Functional Characterization of Dysregulated GEP in NSCLC Tumors

Analysis of the biological and functional significance of the dysregulated GEPs in our responder cases revealed 47 canonical pathways that were significantly different from those of non-responder patients. Among the most commonly differing pathways in the responder cases, we observed a higher level of expression of genes involved in signaling pathways and cellular processes, such as the regulation of immune response, the antigen receptor-mediated signaling pathway, and decreased CD8-positive, alpha-beta T cell, and cytotoxic T cell cytolysis (Table 3).

### 2.3. Validation of the Genomic Signature as a Predicted Response Marker in an Independent Series of Patients Treated with Nivolumab

The value of the signature genes as predictive biomarkers was validated in an independent series of 201 patients, including 22 hepatocellular carcinomas, 14 non-squamous cell lung carcinomas, 5 head and neck squamous cell carcinomas, 1 ureter/renal pelvis carcinoma, 120 melanomas, 4 bladder carcinomas, 31 renal cell carcinomas, and 4 squamous cell lung carcinomas, all treated with nivolumab (https://www.rocplot.org; accessed on 9 November 2023; n = 201; 136 non-responder and 65 responder patients). ROC curves were used to illustrate the expression levels of ITK (AUC: 0.65; 95% CI: 0.62–0.68), NT5E (AUC: 0.64; 95% CI: 0.63–0.65), ITGAL (AUC: 0.63; 95% CI: 0.60–0.65) and CD8A (AUC: 0.60; 95% CI: 0.46–0.76), as the best predictors of response to nivolumab (0.60–0.66; *p* < 0.001), as shown in Figure 2 [22].

### 2.4. Validation of the Clinical Impact of the Genomic Signature in an Independent Series of Patients

In order to explore the clinical outcome of our best predictor biomarkers, we investigated their prognostic effect in an independent series of NSCLC patients from the https://kmplot.com/analysis/index.php?p=service&cancer=lung (accessed on 9 November 2023) database (n = 2166 patients). We found that of the 24 genes that made up our genomic signature, 13 adversely influenced OS. Thus, low levels of expression of CD3G, CXCR6, CTLA3, LCK, ITGAL, ITK, CD73, and HLA-DRB1 and high levels of B2M, NKG5, HLA-A, IFNG, and HLA-B were correlated with extremely short survival (Figure 3). However, the prognostic impact of the expression of the other genes studied could not be confirmed (*p* > 0.05) in this independent series of NSCLC patients. These results indicate that most of the genes identified as biomarkers predicting response are also prognostic factors of the disease.

## 3. Discussion

Gene expression profiling, which interrogates multiple immune-related genes, is increasingly recognized as an attractive approach to characterizing the tumor immune microenvironment and predicting the response to immunotherapy, since it goes further than the measurement of single genes such as PD-L1. In this regard, Ayers et al. [23] used the Nanostring nCounter platform, which is compatible with formalin-fixed paraffin-embedded (FFPE) specimens in the clinical setting, to find evidence that the genomic signature of 18 genes is a reliable tool for identifying pro-inflammatory T-cell phenotypes in NSCLC patients who are likely to respond to PD-L1 inhibition therapy. This technique outperformed PD-L1 immunohistochemistry in PD-L1 non-selected patients. Similarly, Wallden et al. [24] externally validated this genomic signature in an independent patient series, and it is currently being used in several trials with NSCLC patients treated with pembrolizumab (KEYNOTE-189). Recently, Hwang et al. [25] reported that gene expression signatures—specifically of T cells and M1 macrophages—could accurately classify the metastatic NSCLC, without any prior treatment (n = 21), into durable responder and non-durable responder patients [25]. They concluded that levels of expression of CD137 and PSMB9 are better predictors than PD-L1 status or TMB. These studies revealed that GEP is a powerful tool for predicting a response to immunotherapy that outperforms the previously analyzed single biomarkers. The identification of new biomarkers is necessary not only to determine which patients will derive long-term clinical benefit from immunotherapy but also to characterize the mechanisms of acquired resistance to anti-PD-L1 treatment and to progress towards more targeted and precise immunotherapy [26].

Thus, the main clinical practice guidelines establish that the stronger expression of PD-L1 is associated with the stronger response to anti-PD-1/PD-L1 immunotherapy; as a result, the use of PD-1/PD-L1 inhibitors is recommended in the first line when the expression is ≥50% in the tumoral tissue [27]. However, the clinical significance of the response with PD-L1 expression (≥50%) remains uncertain. In fact, in the present study, the three patients with complete response after nivolumab treatment showed ≤ 40% PD-L1 expression (0%, 30%, and 40%). In the second line, benefits have also been observed with immunotherapy combined with chemotherapy when PD-L1 expression is ≥ 1% [28,29,30]. Unfortunately, there is a subgroup of patients who do not benefit from immunotherapy despite having positive PD-L1 expression, while other patients who do not display PD-L1 expression respond adequately to immunotherapy; this is the case for one of the patients in our series, who achieved a complete response after treatment with nivolumab (patient ID4; Table 1) and whose immunohistochemical study revealed an absence of PD-L1 expression. In this regard, Provencio et al. [31] found that PD-L1 staining did not predict long-term survival. Similar results have been obtained in the metastatic setting where PD-L1 has not been shown to predict the outcome of chemotherapy combined with immunotherapy [32]. Specifically, the KEYNOTE-189 trial demonstrated that the addition of pembrolizumab to chemotherapy, as first-line therapy, significantly improved both progression-free survival (PFS) and overall survival (OS) in NSCLC patients with metastatic disease, regardless of their PD-L1 status [33]. Similarly, Rothschild et al. [34] found no significant association between PD-L1 expression levels and the pathological response or nodal downstaging in NSCLC patients treated with neoadjuvant chemotherapy followed by durvalumab. In our study, we identified a significant abnormality in 24 of the 760 genes included in the expression array that allowed us to discriminate between patients with advanced NSCLC responders from non-responders treated with nivolumab. The abnormal expression of some of these genes has previously been reported to predict the response to immunotherapy in different types of neoplasm. For example, a recent study (COLIBRI; phase II) demonstrated the acceptability and safety of a double-immune checkpoint blockade with nivolumab combined with ipilimumab in the neoadjuvant setting, followed by nivolumab in monotherapy as maintenance therapy after chemoradiation in patients with locally advanced squamous cell carcinoma of the cervix. Ray-Coquard et al. [35] found a genomic signature by multivariate analysis when attempting to predict response to treatment that was closely homologous to the one we found in the present study (i.e., aberrant expression of CD8A, CXCL11, CXCL9, CXCR6, GZMA, IDO1, and IFNG genes). Experimentally, it has been shown that neoplastic cells deficient in CXCL9 or CXCL10 are more tumorigenic and have an important role in T-cell infiltration after PD-L1 blockade [36]. Specifically, melanoma patients treated with immunotherapy have been shown to have better survival rates when the tumor expresses CXCL9 and/or CXCL10 [37]. In this regard, our results support the idea that the abnormal expression of CXCL9 and CXCL10 could be good predictors of the response to immunotherapy treatment in patients with NSCLC.

Despite not being able to validate the prognostic value of the genomic signature in our series, due to the small number of patients in our cohort, other studies have shown that these biomarkers are demonstrably useful for predicting disease prognosis. In this regard, the study by Lee et al. [38] showed that IFN-γ-inducible chemokines are useful prognostic markers in patients with NSCLC [38]. They found that the altered expression of CXCL9, CXCL10, CXCL11, and IFN-γ genes were prognostic factors for OS, in univariate analyses, although only overexpression of CXCL9 and CXCL11 were significant independent prognostic factors influencing OS [38,39]. However, it was possible to reverse the adverse prognosis conferred by the abnormal expression of these genes through ICI-based treatments. In a recent investigation involving a functional analysis of CXCL11, Li et al. [40] showed that a high level of expression of this protein is significantly associated with immune-relevant pathways. Consequently, they suggested that CXCL11 expression could behave as a prognostic marker and as a marker of response to immunotherapy treatment in all types of neoplasm.

CD3G and CD8A were other T lymphocyte markers that our study identified as potential predictors associated with both the prognosis and response to ICIs. CD3G and CD8, associated with the T-cell antigen receptor (TCR), contribute to T-cell signaling and their activation, as well as to patient immune response. In fact, the CD8 complex helps recognize and kill virus-infected cells and tumor cells. Previous studies have associated aberrant expression of these genes with OS and the response to ICIs in various neoplasms. In silico studies in uterine cervical squamous cell carcinoma have suggested that elevated CD3G expression could be a novel immunotherapeutic biomarker [41]. A recent paper by Zheng et al. [42] concluded that CD8A may be a useful indicator predicting survival and response to immunotherapy in bladder cancer. The authors analyzed samples from patients with advanced bladder cancer treated with immunotherapy and observed that low levels of CD8A expression were associated with resistance to immunotherapy. However, elevated CD8A expression is associated with a high TMB, with genes involved in the immune checkpoint, and with the presence of several types of tumor-infiltrating immune cells. This is related to the ability to predict a successful response to immunotherapy.

In our genomic signature of 24 prognostically relevant genes, we found high levels of expression of GZMA and HLA-A. In cytotoxic T lymphocyte target cells, Granzyme A activates caspase-independent programmed cell-death pathways that are unique and parallel to those of Granzyme B [43]. In a study of melanoma, Inoue et al. [44] characterized the immune microenvironment of patients before and after treatment with nivolumab and found that patients with high levels of PD-L1, PD-L2, GZMA, and HLA-A expression had a high response rate after immunotherapeutic treatment with nivolumab.

In summary, we report a clear association between the GEP of advanced NSCLC patients and their response to nivolumab that provides a basis for predicting response to ICIs at the time of diagnosis. We also identified new biomarkers that predict the prognosis of the disease. The understanding of differentially dysregulated biological pathways in advanced NSCLC could in future contribute to the design of individualized precision medicine clinical trials based on the reported biomarkers. However, the limited number of cases analyzed so far means that further prospective validation of our genomic signature and prognostic classification is required in larger independent case series involving more patients who are candidates for immunotherapy.

## 4. Materials and Methods

### 4.1. Patients and Samples

In this study, we retrospectively analyzed endoscopic biopsy specimens from 10 patients with advanced non-small cell lung cancer (NSCLC) before they underwent treatment with chemotherapy plus nivolumab as a second line of treatment. Informed consent was obtained from all patients prior to their participation in the study, in accordance with the Declaration of Helsinki. The study was approved by the local ethics committee of the University Hospital of Salamanca (Salamanca, Spain). The tumors were diagnosed and classified according to the criteria of the American Joint Committee on Cancer (AJCC) [45] by a pathologist from the Pathology Department of the University Hospital of Salamanca. Of these patients, 60% were diagnosed with lung adenocarcinoma (ADC), while 40% had squamous cell carcinoma (SCC). OS ranged from 6 to 82 months. The median age was 68 years, with a range of 45 to 80 years. The optimal overall response was assessed and categorized as complete remission (CR), stable disease (SD), or progressive disease (PG) based on the Response Evaluation Criteria in Solid Tumors (RECIST) guidelines, version 1.1 [46]. This evaluation was conducted through computed tomography scans at 6–8-week intervals throughout nivolumab therapy. Individuals exhibiting complete remission (CR) were characterized as “responders” (30% of cases), while those with stable disease (SD) or progressive disease (PG) were designated as “non-responders” (70% of patients). No patient received any additional treatment of nivolumab. Data on clinical (age and sex), associated genetic anomaly, histological (diagnosis and PDL1 staining), and disease-evolutionary (toxicity, type of treatment response, OS, and exitus) variables were collected (Table 1).

### 4.2. RNA Extraction and GEP Studies

Formalin-fixed paraffin-embedded (FFPE) tissue samples were deparaffinized through three washes in Histoclear II (3 min at 50 °C). Subsequently, they were centrifuged at maximum speed for 2 min, and the supernatant was removed. After washing in absolute ethanol twice (maximum speed for 2 min), the sediment was air-dried at room temperature for 10–15 min. Digestion buffer containing 20 mg/mL protease K (Thermo Fisher Scientific Inc., Waltham, MA, USA) was added to each sample and allowed to lyse at 55 °C overnight. The next day, Easy-BLUETM (iNtRON) was used to dissociate the nucleoprotein complexes; chloroform was then added, and the tubes were shaken vigorously for 15 s. The samples were centrifuged at 12,000× *g* for 15 min at 4 °C, and the aqueous phase was transferred to a new tube. The total RNA was precipitated by mixing with isopropyl alcohol. After precipitation, it was stored at −20 °C for a minimum of 1 h and then centrifuged at maximum speed for 10 min at 4 °C.

The RNA pellet was reconstituted in 200 µL DEPC water (Ambion, Austin, TX, USA) and then subjected to precipitation using cold 100% ethanol, 10% sodium acetate, pH 5.2, and 2 µL glycogen 20 mg/mL (Thermo Fisher Scientific) at −20 °C for a minimum of 3 h. Following this, the sample was centrifuged for up to 10 min at 4 °C, then washed with 70% ethanol on ice. Finally, the RNA pellet was dissolved in 10 µL DEPC water.

The Nanodrop platform was employed to assess the quality and quantity of the extracted RNA. The total volume of isolated RNA was treated with DNAseI Amplification Grade (Sigma, St. Louis, MO, USA), and the DNA-free RNA was quantified using a Quant-iT RNA Assay kit (Thermo Fisher Scientific), following the manufacturer’s instructions.

The NanoString nCounter system (nCounter PanCancer IO 360™ Panel; Seattle, WA, USA) was used to study the GEP. This assigns a unique color code to each of the genes under study for subsequent reading by fluorescence. The nCounter Tumor Signaling 360 panel was used, which has the capacity to analyze 760 genes involved in more than 50 signaling pathways related to tumor biology, immune evasion, and microenvironment remodeling.

Starting with the isolated RNA, we adjusted all initial concentrations to achieve a final RNA concentration of 60 ng/µL. After that, the hybridization reaction was prepared, in which the RNA binds to the capture labeling probes by incubation at 65 °C for approximately 18 h. Once the hybridization reaction was completed, the samples were passed to the nCounter Prep Station, where the unbound probe residues were removed, and the labelled sequences were immobilized in the corresponding cartridge, which was then immediately read in the Digital Analyzer to measure the fluorescent label of each of the genes of interest. An RCC file of the level of expression of each gene was then created using the gene read data. In addition to the genes of interest, each experiment also incorporated three control genes (*GAPDH*, *ACTB*, *TUBB*). Raw data were preprocessed and quality-checked using the nanoR package (version 0.1.0) in R (version 4.0.4) (Vienna, Austria). Subsequently, counts were normalized using the housekeeping method based on the expression of the *GAPDH*, *ACTB*, and *TUBB* genes, followed by log_2_ transformation.

Unsupervised multidimensional scaling (MDS) was carried out, and dendrograms were generated using the SIMFIT statistical application (version 7.5.4, Universidad de Salamanca, Salamanca, Spain), taking the Euclidean distance as the distance measure and the group average as the linkage method. In relation to the differential expression analysis, those genes that attained an adjusted value of *p* < 0.05 were considered significant.

### 4.3. Heatmap Studies

Heatmaps were generated using the free webserver Heatmapper (www.heatmapper.ca) [47]. To achieve this, we used the expression data from either 760 genes or the 24 most significant genes obtained from the GEP experiment of patients treated with nivolumab.

### 4.4. In Silico Studies

The online ROC Plotter tool (https://www.rocplot.org, accessed on 9 November 2023) was used to establish a connection between gene expression and therapy response based on transcriptome-level data from cancer patients treated with immunotherapy (anti-PD-1, anti-PD-L1, or anti-CTLA4 therapies), using the Mann–Whitney unpaired U test. With this tool, we validated the 24 genes that showed the most significant deregulation between the non-responder and responder groups of patients treated with nivolumab in our study. This validation was performed at diagnosis in a separate cohort of 201 patients treated with nivolumab (22 hepatocellular carcinomas, 14 non-squamous cell lung carcinomas, 5 head and neck squamous cell carcinomas, 1 ureter/renal pelvis carcinoma, 120 melanomas, 4 bladder carcinomas, 31 renal cell carcinomas, and 4 squamous cell lung carcinomas), consisting of 136 non-responders and 65 responders. The efficiency of the biomarker candidates was assessed by calculating sensitivity, specificity, and the area under the curve (AUC).

The Kaplan–Meier plotter analyses were performed using the online tools available at https://kmplot.com/analysis/, accessed on 9 November 2023. This study aimed to assess the prognostic value of our genomic signature in an independent cohort of NSCLC patients. Patients were categorized into low- or high-expression level groups through median dichotomization. OS was used as the endpoint for survival assessment. Probes used for each gene were 206804_at (CD3G), 206974_at (CXCR6), 205488_at (CTLA3), 204891_s_at (LCK), 213475_s_at (ITGAL), 211339_s_at (ITK), 203939_at (CD73), 204670_x_at (HLA-DRB1), 216231_s_at (B2M), 205495_s_at (NKG5), 215313_x_at (HLA-A), 210354_at (IFNG), and 209140_x_at (HLA-B) in 2166 patients.

Gene enrichment analysis was performed using the publicly accessible EnrichR online platform (https://maayanlab.cloud/Enrichr/#, accessed on 9 November 2023). For this analysis, we specifically chose the 24 genes that exhibited the most significant deregulation in our study, distinguishing between nivolumab-treatment responder and non-responder groups of patients. The aim was to identify the biological processes and pathways associated with our selected set of genes.

### 4.5. Statistical Methods

The means, standard deviations (SDs), and ranges of continuous variables, as well as the frequencies and percentages of dichotomous variables, were calculated using IBM SPSS for Windows version 20.0 (IBM Corp., Armonk, NY, USA).

## Figures and Tables

**Figure 1 ijms-25-02048-f001:**
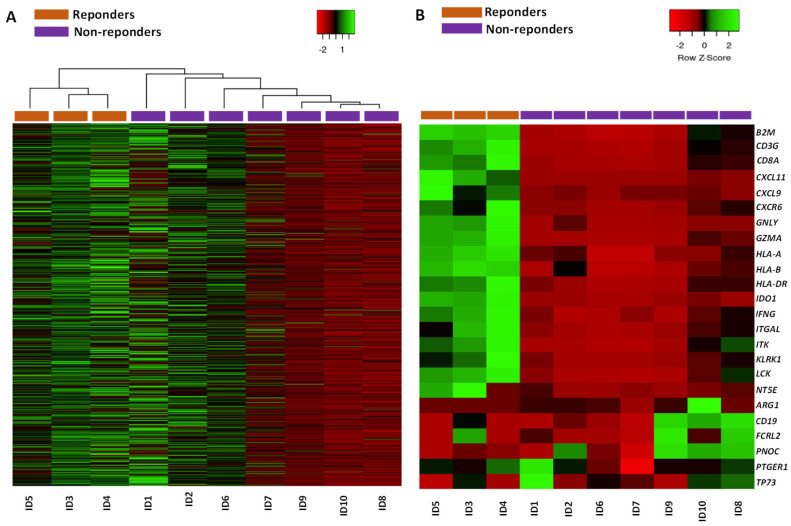
Association between non-small cell lung carcinoma (NSCLC) tumor-specific gene expression profiles (GEPs) and the response to nivolumab treatment. Unsupervised hierarchical clustering analysis and the corresponding GEP heatmap reveal clear differences in the profile between responder and non-responder NSCLC patients based on the combination of expression of 760 genes (**A**); nCounter^®^ Tumor Signaling 360 Panel) and the 24 individual genes (**B**) selected by the prediction algorithms, which are better at discriminating between NSCLC patients who do and do not respond to nivolumab treatment.

**Figure 2 ijms-25-02048-f002:**
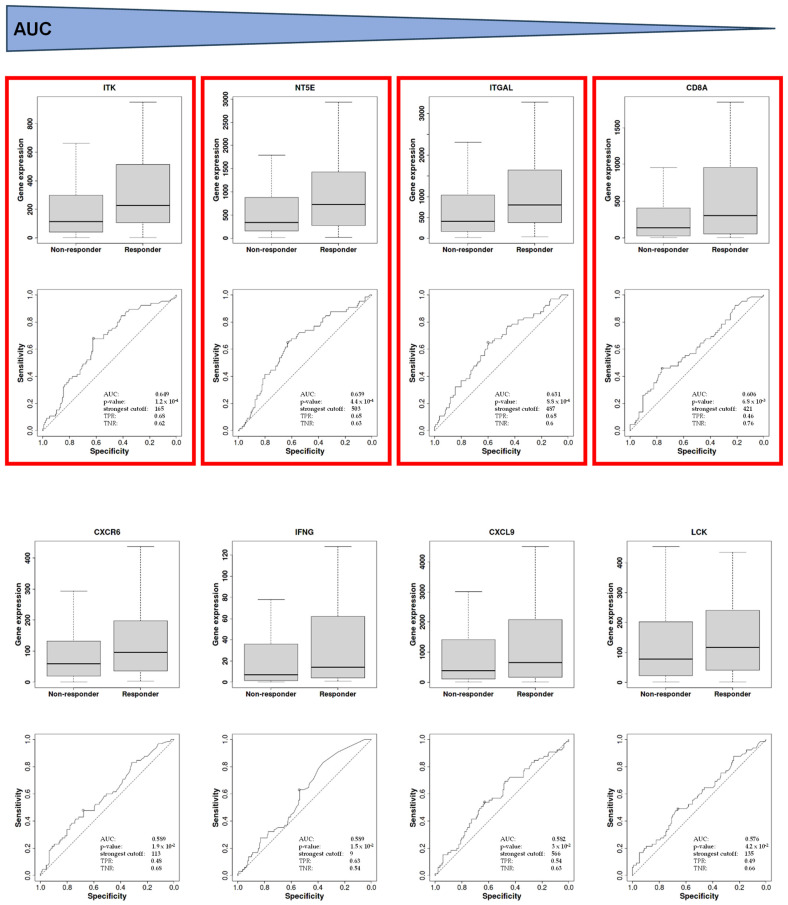
Bar charts showing statistically significant differences in the expression levels of ITK, NT5E, ITGAL, CD8A, CXCR6, IFNG, CXCL9, and LCK genes between patients responding and not responding to nivolumab treatment (*p* < 0.001). The expression levels and ROC curves were obtained from public databases (https://www.rocplot.org, accessed on 9 November 2023) containing genomic data obtained through RNAseq techniques from a total of 201 patients treated with nivolumab (136 non-responders and 65 responders). The bar charts are ordered, from highest to lowest, based on the AUC (area under the curve). The red boxes indicate the genes that exhibited an AUC value above 0.6. The triangle at the top of the figure indicates that the genes are arranged from the highest to the lowest AUC value.

**Figure 3 ijms-25-02048-f003:**
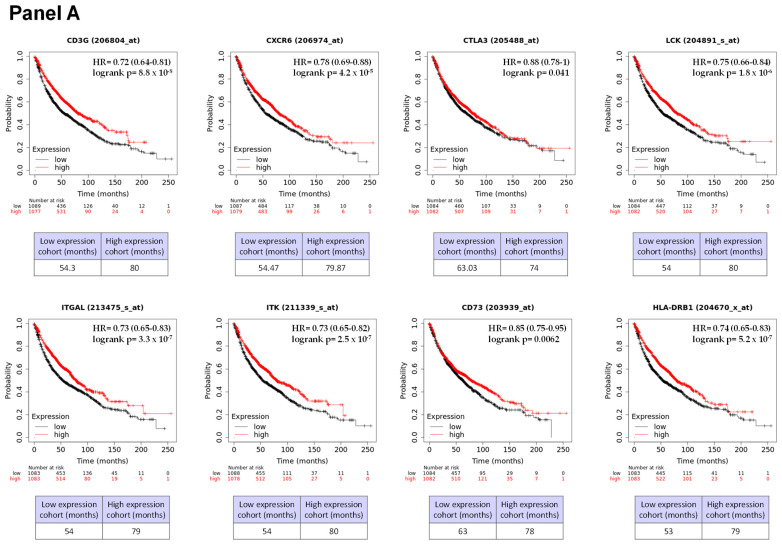
Impact of the individual expression of each gene comprising our genomic signature on overall survival (OS) in an independent series of NSCLC patients from https://kmplot.com/analysis/index.php?p=service&cancer=lung database (n = 2166 patients). A small majority of the genes (13/24; 54%) that make up our genomic signature had a significant influence on OS, as indicated by the multivariate analysis. Low levels of expression of CD3G, CXCR6, CTLA3, LCK, ITGAL, ITK, CD73, and HLA-DRB1 (**A**) and high levels of expression of B2M, NKG5, HLA-A, IFNG, and HLA-B (**B**) were associated with extremely short survival.

**Table 1 ijms-25-02048-t001:** Clinico-biological and pathological characteristics of 10 patients with stage IV non-small cell lung carcinoma (NSCLC) treated with nivolumab.

ID	Gender	Age (Years)	%PDL1 Expression *	Genetic Alterations	Histological Diagnosis	Response to Nivolumab	OS (Months)	Exitus	Toxicity
**1**	M	74	1	ND	Adenocarcinoma	PG	6	yes	Asthenia
**2**	M	54	5	ND	Adenocarcinoma	PG	37	yes	No
**3**	M	67	30	ND	Squamous	CR	82	no	No
**4**	M	68	0	NA	Squamous	CR	65	no	Asthenia
**5**	M	73	40	NA	Adenocarcinoma	CR	74	no	No
**6**	F	76	80	ND	Squamous	PG	15	yes	No
**7**	M	80	1	ND	Adenocarcinoma	PG	25	yes	Arthralgia
**8**	M	78	30	ND	Adenocarcinoma	PG	20	yes	No
**9**	M	45	50	*EGFR* mutations (T790M and exon 19)	Adenocarcinoma	PG	37	yes	No
**10**	M	62	5	ND	Squamous	SD	22	yes	No

M: male; F: female; ND: not detected; * Clone 22C3 was used to detect PD-L1 by immunochemistry; PG: progression; SD: stabilized disease; NA: not analyzed; CR: complete remission; OS: overall survival.

**Table 2 ijms-25-02048-t002:** Statistically significant deregulated mRNA transcript genes in responder (n = 3) vs. non-responder (n = 7) patients with non-small cell lung carcinoma (NSCLC) treated with nivolumab.

Gene Name	Gene ID	Responder vs. Non-Responder Patients (n-Fold Change)	Chromosomal Band	*p*
Up-regulated transcripts in responders vs. non-responders NSCLC				
*CXCL11*	NM_6373	5.2	4q21.1	0.007
*NT5E*	NM_4907	3.2	6q14.3	0.02
*KLRK1*	NM_22914	3.2	12p13.2	0.04
*CD3G*	NM_917	3.1	11q23.3	0.04
*GZMA*	NM_3001	2.9	5q11.2	0.01
*IDO1*	NM_3620	2.8	8p11.2	0.01
*LCK*	NM_3932	2.7	1p35.2	0.05
*CXCL9*	NM_4283	2.7	4q21.1	0.05
*GNLY*	NM_10578	2.6	2p11.2	0.03
*ITGAL*	NM_3628	2.5	16p11.2	0.05
*HLA-DRB1*	NM_3123	2.0	6p21.3	0.05
*CXCR6*	NM_10663	2.1	3p21.3	0.05
*IFNG*	NM_3458	2.1	12q15	0.05
*CD8A*	NM_925	2.1	2p11.2	0.05
*ITK*	NM_3702	1.9	5q33.3	0.05
*B2M*	NM_567	1.8	15q21-1	0.02
*HLA-B*	NM_3106	1.2	6p21.3	0.05
*HLA-A*	NM_3105	0.7	6p22.1	0.05
Down-regulated transcripts in responders vs. non-responders NSCLC				
*PNOC*	NM_5368	−4.5	8p21.1	0.01
*CD19*	NM_930	−4.3	16p11.2	0.04
*TP73*	NM_7161	−3.7	1p36.3	0.02
*ARG1*	NM_9439	−3.7	6q23.2	0.04
*FCRL2*	NM_79368	−3.6	1q23.1	0.03
*PTGER1*	NM_5731	−3.3	19p13.1	0.04

**Table 3 ijms-25-02048-t003:** Most representative canonical pathways involved in NSCLC tumors as defined by the gene expression profile (GEP) of the most strongly dysregulated transcripts in responder compared with non-responder patients. Table of data displaying the term, *p*-value, *q*-value, *z*-score, or combined score from the Gene Enrichment analysis of the 24 genes that exhibited the most significant dysregulation in our study.

Term	Library	*p*-Value	q-Value	z-Score	Combined Score
Regulation of immune response (GO:0050776)	GO_Biological_Process_2021	3.5048 × 10^−9^	1.7734 × 10^−6^	699	2005
Antigen processing and presentation	KEGG_2021_Human	3.6807 × 10^−6^	0.00026869	9215	2002
Epstein-Barr virus infection	KEGG_2021_Human	0.000028904	0.0001055	4177	8213
Human T-cell leukemia virus 1 infection	KEGG_2021_Human	0.000050659	0.0012327	3839	7332
Natural killer cell mediated cytotoxicity	KEGG_2021_Human	0.000085964	0.0015688	5294	9831
Interferon-gamma-mediated signaling pathway (GO:0060333)	GO_Biological_Process_2021	0.00015811	40,002	8318	1494
Decreased CD8-positive, alpha-beta T cell number MP:0008079	MGI_Mammalian_Phenotype_Level_4_2021	0.00021816	83,338	4496	7932
Decreased cytotoxic T cell cytolysis MP:0005079	MGI_Mammalian_Phenotype_Level_4_2021	0.00056055	10,707	1425	2379
Antigen receptor-mediated signaling pathway (GO:0050851)	GO_Biological_Process_2021	0.00067675	92,628	3687	6086
Antigen processing and presentation of exogenous peptide antigen via MHC class I, TAP-independent (GO:0002480)	GO_Biological_Process_2021	0.00084676	92,628	5706	9292
Positive regulation of T cell mediated immunity (GO:0002711)	GO_Biological_Process_2021	0.00009153	92,628	1247	2020
Allograft rejection	KEGG_2021_Human	0.0011451	16,494	1173	1875
Abnormal cytotoxic T cell physiology MP:0005078	MGI_Mammalian_Phenotype_Level_4_2021	0.0014155	18,025	1108	1747
Abnormal T cell activation MP:0001828	MGI_Mammalian_Phenotype_Level_4_2021	12,335	1178	6223	8466
Decreased susceptibility to autoimmune diabetes MP:0004804	MGI_Mammalian_Phenotype_Level_4_2021	60,339	429	1056	1269

## Data Availability

The data presented in this study are available at https://doi.org/10.7910/DVN/IZTAPH (accessed on 22 December 2023).

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
