# Peer review of "Differentially Expressed Genes Involved in Primary Resistance to Immunotherapy in Patients with Advanced-Stage Pulmonary Cancer"

_ijms, 2024, doi:10.3390/ijms25042048_

Round 1

Reviewer 1 Report

Comments and Suggestions for Authors

The authors present a panel of potential prognostic biomarker panel that could be used in therapeutic decision making in regards to administering ICB in particular nivolumab to patients with advanced non-small cell lung cancer (NSCLC) patients. There are some major issues the study findings presented:

1. The authors fail to mention that the results shown in results section 2.3 of the independent series of 254 patients treated with nivolumab is comprised primarily of data collected from patients with other cancers. Only 14 of these patients have NSCLC. It’s not clear why this point was omitted and raises some concern about the validation of the study findings.

2.    Also, as the data from here is being compared to pre-treated patient samples from the original cohort (n=10), why has data from patients ‘ongoing’ treatment also been included in the nivolumab treated patients?  The authors should omit those patients.

3.       The relevance of section 2.4 is unclear and seems to almost contradict the data shown in sections 2.1 and 2.5. Moreover, line 168-169 – how did the authors determine the ‘same’ result in seven of the ten patients?

4.       Did the authors find correlations between the 13 genes found to predict shorter overall survival (OS) from the large NSCLC (2,166 cases) with their own cohort – particularly among the non-responders with progressive disease? If not, please consider including this analyses/discussion.

5.       Did the authors monitor the changes in the 24 genes in pre- and post-treated samples?

6.       IHC expression profiles of top genes found to be differentially expressed in responder and non-responder patients samples could enhance the significance of the results presented.

7.       Were the non-responders put on any other ICB over the course of their disease progression?

Author Response

Comment 1.- The authors present a panel of potential prognostic biomarker panel that could be used in therapeutic decision making in regards to administering ICB in particular nivolumab to patients with advanced non-small cell lung cancer (NSCLC) patients. There are some major issues the study findings presented: 1. The authors fail to mention that the results shown in results section 2.3 of the independent series of 254 patients treated with nivolumab is comprised primarily of data collected from patients with other cancers. Only 14 of these patients have NSCLC. It’s not clear why this point was omitted and raises some concern about the validation of the study findings.

Answer to comment 1.- The number and tumor type of the 254 patients treated with nivolumab are now clearly described in the material and methods section of the revised manuscript (now 201 cases, see comment 2).

Comment 2.- Also, as the data from here is being compared to pre-treated patient samples from the original cohort (n=10), why has data from patients ‘ongoing’ treatment also been included in the nivolumab treated patients?  The authors should omit those patients.

Answer to comment 2.- Following the reviewer's suggestions, we have eliminated the data from patients “ongoing” treatment that was initially evaluated in the online ROC Plotter, leaving only the data from pretreated patients. The results of the validation of our genetic signature of predictive biomarkers performed in these pre-treated patients are shown in the new figure 3. Additionally, these updated findings have been incorporated throughout the entire manuscript.

Comment 3.- The relevance of section 2.4 is unclear and seems to almost contradict the data shown in sections 2.1 and 2.5. Moreover, line 168-169 – how did the authors determine the ‘same’ result in seven of the ten patients?

Answer to comment 3.- Both reviewers have identified difficulties in the interpretation of the results of section 2.4. Because of this, we have removed this section (methodology, results, and figure 4) since the results it provides are not essential to the main purpose of the manuscript.

Comment 4.- Did the authors find correlations between the 13 genes found to predict shorter overall survival (OS) from the large NSCLC (2,166 cases) with their own cohort – particularly among the non-responders with progressive disease? If not, please consider including this analyses/discussion.

Answer to comment 4.- In accordance with the reviewer's suggestion, we have included a paragraph in the discussion section, indicating that due to the small number of patients included in our own cohort, we were unable to validate the correlation between the expression levels of the 13 genes and survival of the disease.

Comment 5.- Did the authors monitor the changes in the 24 genes in pre- and post-treated samples?

Answer to comment 5.- As several patients achieved a complete response, we have not been able to evaluate the changes in the expression levels of the 24 genes in the post-treatment samples.

Comment 6.- IHC expression profiles of top genes found to be differentially expressed in responder and non-responder patients samples could enhance the significance of the results presented.

Answer to comment 6.- We fully agree with the reviewer´s comment. In fact, we are currently conducting this study, but unfortunately, we do not have results available to include in this study yet.

Comment 7.- Were the non-responders put on any other ICB over the course of their disease progression?

Answer to comment 7.- The patients studied did not receive any other additional treatment. This has been clarified in the materials and methods section of the revised manuscript.

Reviewer 2 Report

Comments and Suggestions for Authors

In this study, the authors reported that gene expression was different between the two groups of patients treated with nivolumab, using three patients who responded and seven patients who did not respond, and performing immune-related gene expression analysis.

The means and methods, chart preparation, and explanations of the figures are all difficult to understand and need to be reviewed. In particular, the purpose of the analysis using a data set unrelated to the therapeutic effect of immune checkpoint inhibitors and its interpretation is not clear.

Point out some of them.

Table 1. What is the definition of responder and non-responder?

What antibodies for PD-L1 were used, what are the outcomes for OS, and what are the units of OS? Are all patients dying? What is the PFS? What is the treatment line for nivolumab? Were biopsy specimens taken just before nivolumab? What was the response determined by? RECIST? What are the definitions of PG and SD?

Figure 2. Where is the description of the method? There are no units, but what does the length of the bar represent, and where is the cutoff for determining increased and decreased expression?

Figure 3. Can you make the method of describing p-values clearer? What does the red enclosure mean, showing AUC of 0.6 or higher, describe it in the figure legend. What is the meaning of the triangle? Does it simply mean they are arranged in order of AUC from left to right? Does the height of the triangle mean anything?

The contents of Figures 4 and 5 are based on data sets that have nothing to do with the therapeutic effects of immune checkpoint inhibitors, and I do not understand the intention behind the analysis and the interpretation of the results. In particular, looking at the results in Figure 5, does it not mean that many of the factors extracted in the 10 cases are simply looking at prognostic factors rather than predictors of treatment efficacy?

Figure 4 Panel B. The numbers on the vertical axis are listed only on the upper left, but what about the others? It isn't easy to understand the boundary between each biomarker.

Paragraphs 1-3 of the discussion are background. Shouldn't we start with the contents of paragraph 4?

Author Response

Comment 1.- In this study, the authors reported that gene expression was different between the two groups of patients treated with nivolumab, using three patients who responded and seven patients who did not respond, and performing immune-related gene expression analysis.

The means and methods, chart preparation, and explanations of the figures are all difficult to understand and need to be reviewed. In particular, the purpose of the analysis using a data set unrelated to the therapeutic effect of immune checkpoint inhibitors and its interpretation is not clear.

Table 1. What is the definition of responder and non-responder? What antibodies for PD-L1 were used, what are the outcomes for OS, and what are the units of OS? Are all patients dying? What is the PFS? What is the treatment line for nivolumab? Were biopsy specimens taken just before nivolumab? What was the response determined by? RECIST? What are the definitions of PG and SD?

Answer to comment 1.- The material and methods, preparation of graphs, and explanations of the figures have been reviewed. The answers to the questions raised by the reviewer have been added in the figure legend of table 1 and material and methods section (definition of responder and non-responder, antibody used to detect PD-L1, etc.)

Comment 2.- Figure 2. Where is the description of the method? There are no units, but what does the length of the bar represent, and where is the cutoff for determining increased and decreased expression?

Answer to comment 2.- The method corresponding to Figure 2 is described on page 12, lines 384-392. Following the Reviewer's instructions, we have added more information regarding this graph in the material and methods section. Additionally, we have included a supplementary table indicating the terms, library, p-value, q-value, z-score, and combined score corresponding to Figure 2.

Comment 3.- Figure 3. Can you make the method of describing p-values clearer? What does the red enclosure mean, showing AUC of 0.6 or higher, describe it in the figure legend. What is the meaning of the triangle? Does it simply mean they are arranged in order of AUC from left to right? Does the height of the triangle mean anything?

Answer to comment 3.- In the materials and methods section, we have included the statistical test used by the online ROC Plotter tool. As indicated by the Reviewer, the red boxes are those genes that showed an AUC value above 0.6 Moreover, the triangle at the top of the figure indicates that the genes are placed from highest to lowest AUC value. We have added information that is now included in the figure legends.

Comment 4.- The contents of Figures 4 and 5 are based on data sets that have nothing to do with the therapeutic effects of immune checkpoint inhibitors, and I do not understand the intention behind the analysis and the interpretation of the results. In particular, looking at the results in Figure 5, does it not mean that many of the factors extracted in the 10 cases are simply looking at prognostic factors rather than predictors of treatment efficacy?

Answer to comment 4.- Both reviewers have identified difficulties in the interpretation of the figure 4. Because of this, we have removed this figure since the results it provides are not essential to the main message of the study. Furthermore, we have added a sentence in the results section (section 2.4) of the new revised version of the manuscript showing that some of the biomarkers found to be the best predictors of treatment (section 2.3), are also prognostic factors of the disease.

Comment 5.- Figure 4 Panel B. The numbers on the vertical axis are listed only on the upper left, but what about the others? It isn't easy to understand the boundary between each biomarker. 

Answer to comment 5.- Figure 4 has been removed in the revised version of the manuscript.

Comment 6.- Paragraphs 1-3 of the discussion are background. Shouldn't we start with the contents of paragraph 4?

Answer to comment 6.- Following the Reviewer's instructions, we have removed paragraphs 1-3 of the discussion. Paragraph 2 has been restructured and we have started the discussion with paragraph 4.

Round 2

Reviewer 2 Report

Comments and Suggestions for Authors

Thank you for posting the revised version.

Line 376-397: The authors have added additional criteria for response and non-response, but I think "pathological remission" is incorrect.

Supplement Table 1 was added, which is understandable, but Fig. 2 is a bar chart specific to software using analysis software, which I think is difficult to understand.

Author Response

Comment 1.- Line 376-397: The authors have added additional criteria for response and non-response, but I think "pathological remission" is incorrect.

Answer to comment 1.- We thank the reviewer for noting this mistake that has now been corrected. Line 379-389 has been rewritten adding additional criteria for response and non-response. Additionally, a reference to this has been added.

Comment 2.- Supplement Table 1 was added, which is understandable, but Fig. 2 is a bar chart specific to software using analysis software, which I think is difficult to understand

Answer to comment 2.- Following the reviewer's suggestions, Figure 2 has been removed, and in its place, the information from Supplementary Table 1 has been placed, which is now named Table 3.
